# Shape Dependence of Silver-Nanoparticle-Mediated Synthesis of Gold Nanoclusters with Small Molecules as Capping Ligands

**DOI:** 10.3390/nano13162338

**Published:** 2023-08-14

**Authors:** Cheng-Yeh Chang, Yi-Ru Wu, Tzu-Hsien Tseng, Jun-Hao Su, Yu-Shan Wang, Fang-Yi Jen, Bo-Ru Chen, Cheng-Liang Huang, Jui-Chang Chen

**Affiliations:** Department of Applied Chemistry, National Chiayi University, Chiayi City 600355, Taiwan; sss1072727@gmail.com (C.-Y.C.); jerry19970228@gmail.com (Y.-R.W.); tzuhome50@gmail.com (T.-H.T.); andsonhao@gmail.com (J.-H.S.); wang30534@gmail.com (Y.-S.W.); sabrinajengg@gmail.com (F.-Y.J.); s1090251@mail.ncyu.edu.tw (B.-R.C.); clhuang@mail.ncyu.edu.tw (C.-L.H.)

**Keywords:** gold nanoclusters, silver nanoparticles, quantum yields, fluorescence, TEM

## Abstract

In this study, differently shaped silver nanoparticles used for the synthesis of gold nanoclusters with small capping ligands were demonstrated. Silver nanoparticles provide a reaction platform that plays dual roles in the formation of Au NCs. One is to reduce gold ions and the other is to attract capping ligands to the surface of nanoparticles. The binding of capping ligands to the AgNP surface creates a restricted space on the surface while gold ions are being reduced by the particles. Four different shapes of AgNPs were prepared and used to examine whether or not this approach is dependent on the morphology of AgNPs. Quasi-spherical AgNPs and silver nanoplates showed excellent results when they were used to synthesize Au NCs. Spherical AgNPs and triangular nanoplates exhibited limited synthesis of Au NCs. TEM images demonstrated that Au NCs were transiently assembled on the surface of silver nanoparticles in the method. The formation of Au NCs was observed on the whole surface of the QS-AgNPs if the synthesis of Au NCs was mediated by QS-AgNPs. In contrast, formation of Au NCs was only observed on the edges and corners of AgNPts if the synthesis of Au NCs was mediated by AgNPts. All of the synthesized Au NCs emitted bright red fluorescence under UV-box irradiation. The synthesized Au NCs displayed similar fluorescent properties, including quantum yields and excitation and emission wavelengths.

## 1. Introduction

Over the past decades, the nanoscience of noble metals has drawn tremendous attention and become a fascinating research subject. Amongst the noble metals, gold is considered as one of the most extensively studied elements, and a myriad of related articles have been produced due to its chemical stability, specific photoproperties, nontoxicity, and biocompatibility. It is involved in various research areas and applications, including chemical sensing [1], medicine [2], and catalysis [3].

Gold exhibits characteristic properties that are influenced by its particle size. When the particles are prepared to be 100 nm or less in diameter, they are termed “nanomaterials”. These small-sized gold nanomaterials are further classified into two different groups based on their properties and/or dimensions, which are gold nanoparticles (AuNPs) and gold nanoclusters (Au NCs). Au NCs usually consist of several to tens of gold atoms. Structurally, gold clusters are divided into two parts, the outer shell and the inner core. The outer shell is composed of protecting ligands or capping ligands coordinated with the inner core, the aggregate of gold atoms. The inner core of the cluster has to be segregated by the capping ligands to prevent further aggregation into larger nanoparticles. The sizes of gold nanoclusters are typically less than 5 nm in diameter, exhibiting intrinsically unique and size-specific properties. The ultra-small nanoclusters are in a group of their own, as their unique properties are much different from those of Au nanoparticles. For instance, AuNPs absorb visible wavelengths, but Au NCs do not. Unlike AuNPs, Au NCs neither appear in any specific geometrical shape under the detection of TEM or SEM, nor do they display surface plasmon resonance (SPR) under UV-Vis spectroscopy [4]. The most important characteristics distinguishing Au NCs from AuNPs is that the smaller Au NCs exhibit a molecular-like luminescence property [2,5]. This unique phenomenon is attributed to the size of these nanoclusters, similar to an electron Fermi wavelength of gold metal, leading to its continuous band structure, which becomes quantum-like and breaks into discrete energy states [5]. As a result, these Au NCs behave like molecules and emit fluorescence even though there are no conjugated double bonds in a rigid ring [6].

The preparation method of Au NCs is classified into two approaches, “top-down” and “bottom up”. Both synthetic procedures require the protection of capping ligands during synthesis. The top-down approach synthesizes smaller-sized Au NCs from larger-sized nanoparticles assisted with itching reagents and solvents [1,7,8]. However, this approach is time-consuming, and isolation is typically required [9]. It also requires large nanoparticles as the precursors, which limits its further development. The “bottom-up” approach usually prepares Au clusters through the reduction of Au ions into atoms, followed by the spontaneous nucleation of the Au atoms. However, Au ions can be reduced into Au atoms and then form nanoparticles, instead of nanoclusters, because of the potential aggregation of Au atoms in the absence of capping ligands. Too many factors have to be taken into consideration for the successful synthesis of Au NCs. In general, the reaction condition for each synthesis of Au NCs is always dependent on the properties of the capping ligands. Selecting suitable capping ligands has become a challenge for the “bottom-up” synthesis of Au NCs. In the past two decades, the synthesis of Au NCs using macromolecules as the capping ligands has attracted the most attention, because macromolecules are capable of sequestering limited Au ions in a restricted space so that fewer Au nanoparticles are produced. Researchers have used dendrimers and proteins as capping ligands for the preparation of Au NCs [10,11,12,13]. However, applications of Au NCs are always dictated by the properties or functions of capping ligands. The synthesis of Au NCs with small molecules is frequently required. For instance, Au NCs synthesized with cysteamine or 4-aminothiophenol can associate with monoamine oxidase B (MAO) and inhibit its activity (MAOI). These MAOI-like Au NCs can mimic drugs to treat patients with depression or anxiety [14]. Siddiqui et al. synthesized GST-Au NCs and utilized fluorescence resonance energy transfer (FRET) to detect melanin. The phenomenon of FRET was achieved by bringing melanin into the proximity of GST-Au NCs through the interaction of GST with melanin [15]. However, Au NCs with small molecules as capping agents are more difficult to synthesize than those with polymer molecules as capping ligands. There is no facile method that can be generally adopted for the synthesis of Au NCs with small molecules, although several small capping ligands have been used to synthesize Au NCs, such as glutathione disulfide (GSSG) [16]. However, this chemical reduction method is time-consuming (about 2 days) and produces unwanted large nanoparticles as byproducts. Several researchers have also reported the synthesis of GST-Au NCs [17,18,19]. Nevertheless, the synthesis process requires high temperatures or further purification. 

Our lab reported a new synthesis approach that could be applied to the synthesis of fluorescent Au NCs using small molecules as capping ligands [20]. The synthesis of Au NCs was completed under mild conditions within one day at 37 °C when quasi-spherical AgNPs were available. No Au nanoparticles were produced as byproducts after Au NC synthesis. The new method was demonstrated to synthesize Au NCs with two different molecules as the capping ligands, GSSG and the reduced form of DTT. As we previously suggested, quasi-spherical AgNPs play two important roles in Au NC synthesis: they act as a reducing agent for gold ions and serve as a platform to accumulate capping ligands on the surface of AgNPs. Au ions on the surface of quasi-spherical AgNPs are trapped by capping ligands and reduced by the AgNPs, while the capping ligands are attracted to the surface of the AgNPs. The assembly of Au ions and capping ligands on the surface of AgNPs turns on the synthesis of Au NCs. One advantage of this method is that the capping ligands can be variable if they can associate with AgNPs. However, the morphology of the AgNPs could influence their ability to attract capping ligands to the surface. Thus, it is not clear whether or not the shape of AgNPs is critical for their synthesis. In this report, four different shapes of AgNPs, quasi-spherical silver nanoparticles (QS-AgNPs), spherical silver nanoparticles (S-AgNPs), triangular silver nanoplates (T-AgNPts) and silver nanoplates (AgNPts), were prepared and used in the experiments to examine the effect of AgNP morphologies on the synthesis of Au NCs. Three small molecules, DTT, oxidized DTT (DTTox), and GSSG were used to synthesize Au NCs as capping ligands to examine whether the synthesis of Au NCs is dependent of the shape of the AgNPs. 

## 2. Materials and Methods

### 2.1. Materials 

Chloroauric acid trihydrate (HAuCl_4_•3H_2_O) (99.99%) was obtained from Alpha Aesar (Thermo Fisher Scientific, Ward Hill, MA, USA). Sodium citrate dihydrate (C_6_H_5_Na_3_O_7_•2H_2_O) (>99.0%) was purchased from J. T. Baker (J.T. Baker, Philipsburg, NJ, USA). DL, 1, 4-dithiothreitol (DTT) (99%) was purchased from Acros Organics (Geel, Belgium). Silver nitrate (≥99%), glutathione disulfide (GSSG) (≥99%), sodium borohydride (99%), and oxidized DTT (≥98%) were all purchased from Sigma Aldrich (Sigma Aldrich Inc. St. Louis, MO, USA).

### 2.2. Synthesis of AgNPs

#### 2.2.1. Synthesis of Quasi-Spherical AgNPs

The procedure for synthesizing quasi-spherical AgNPs followed the method described previously, except that the violet LEDs (the average intensity 40 mW/cm^2^ and wavelength 405 nm) were used to replace the UVB lamp in the irradiation chamber for the reactions [21]. In general, a reaction bottle containing 40 mL ddH_2_O was mixed with the silver nitrate solution (0.5 mL, 0.01 M). Sodium citrate (0.066 g) was added to the solution and mixed thoroughly. The reaction solution was added with ddH_2_O to a total of 50 mL to reach the final concentrations of 0.1 mM AgNO_3_ and 4.5 mM sodium citrate. The reaction solution was then irradiated under violet LEDs in the chamber for 2.5 h (Appendix A).

#### 2.2.2. Synthesis of Spherical AgNP

In brief, silver nitrate solution (0.5 mL, 0.01 M) and sodium citrate (0.1 mL, 0.03 M) were sequentially added into a beaker containing 48.5 mL ddH_2_O and mixed thoroughly. Freshly prepared sodium borohydride (0.5mL, 0.01 M) was then placed into the solution. Double distilled H_2_O was placed into the solution to make a total of 50 mL to obtain final concentrations of 0.1 mM AgNO_3_, 0.06 mM sodium citrate, and 0.1 mM NaBH_4_. The solution was then mixed with stirring for 10 min. The mixture was then irradiated under violet LEDs (405 nm) in the chamber for 80 min.

#### 2.2.3. Synthesis of Silver Nanoplates (AgNPts)

The steps used for the synthesis of silver nanoplates followed the procedure that was previously reported [22]. In brief, the synthesis of AgNPts begins with the same protocol as the preparation of quasi-spherical AgNPs. The quasi-spherical AgNP colloid solution was used to undergo shape transformation to form AgNPts by the following steps. The quasi-spherical AgNPs were irradiated under a green LED set-up in a chamber (*λ_max_* 520 nm, average power 48 mW/cm^2^) for 2 h. The solution was then changed to another chamber for the irradiation of red LEDs (620 ± 18 nm, average power 116 mW/cm^2^) for 3 h (Appendix A). The solution was then removed from the LED chamber to stop the reaction. A UV-Visible spectrophotometer was used to confirm whether the surface plasmon resonance absorption band was formed at approximately 700 nm, indicating the synthesis of silver nanoplates.

#### 2.2.4. Synthesis of Triangular Nanoplates (T-AgNPts)

We prepared a solution (50 mL) containing the final concentrations of 0.1 mM AgNO_3_, 4.5 mM sodium citrate, and 0.1 mM NaBH_4_. The solution was mixed by stirring for 5 min. The mixture was then irradiated under sodium LEDs (589 nm) for 1.5 h. A UV-Visible spectrophotometer was employed to verify whether the surface plasmon resonance absorption band was observed at approximately 700 nm, an indication that silver triangular nanoplates had been synthesized.

### 2.3. Synthesis of Au NCs

Typically, a reaction aqueous solution (1 mL) was formed with final concentrations of HAuCl_4_ and capping ligands of 0.4 mM and 15 mM, respectively. The total silver concentration of AgNPs was 4.8 × 10^−2^ mM. The final concentration of the capping ligands may be changed, and it will be specified in the discussion. The solution was then incubated at 37 °C for 7 h or longer. The reaction solution was monitored under a UV-box and a fluorescent spectrophotometer. Dialysis was employed to remove silver ions using a 2.0 kDa MWCO dialysis bag, if necessary.

### 2.4. Fluorescence Spectroscopy

For a typical fluorescent measurement, a quartz cuvette (4 mm × 4 mm) was used and placed with fluorescent samples (250 µL). A Teflon-coated magnetic stir flea (2 mm × 2 mm) was used to facilitate mixing, if necessary. Each fluorescence sample was measured by a Fluorolog^®^-3 spectrophotometer (HORIBA Jobin Yvon Inc, Edison, NJ, USA). The temperature of the sample chamber was maintained at room temperature. Each measurement was subtracted by a blank sample (ddH_2_O). The excitation shutter was kept closed, except during measurements, to avoid the continuous irradiation of light.

### 2.5. TEM Imaging of AgNPs and Au NCs

A transmission electron microscope (TEM), Joel JEM-2100 (purchased from Japan Electron Optics Laboratory Co., Ltd., Tokyo, Japan), was used to detect Au NCs and AgNPs. The TEM instrument was operated under air-conditioning at 100 kV and 200 kV. To prepare samples for the TEM detection, the as-prepared Au NCs or AgNP colloidal solutions were placed onto a carbon-coated copper grid. The samples were then air-dried at ambient temperature before examination.

### 2.6. UV-Visible Spectroscopy

The UV-Vis extinction spectra of all samples were measured at room temperature by the Hitachi U-5100 spectrophotometer (Hitachi Science & Technology, Tokyo, Japan) using a quartz cuvette (1.0 cm × 1.0 cm).

### 2.7. Quantum Yields

The quantum yield was calculated to estimate the fluorescent efficiency of a newly synthesized Au NC sample. It was defined and calculated as follows:

Quantum yield = number of protons emitted/number of protons absorbed, or
Q_S_ = Q_R_(I_S_ × OD_R_ × n_S_^2^)/(I_R_ × OD_S_ × n_R_^2^)(1)
where Q_S_ is the sample’s calculated quantum yield; Q_R_ is the standard’s quantum yield (ethidium bromide, 20%); I_S_ is the sample’s total fluorescent intensity; I_R_ is the standard’s total integrated fluorescent intensity (ethidium bromide, EtBr); OD_S_ is the optical density for the sample at the absorption (or excitation) wavelength; OD_R_ is the optical density for EtBr as the reference at the absorption wavelength; n_S_ is the refraction index of the sample solution; and n_R_ is the refraction index of the EtBr solution. The values of n_S_ and n_R_ were very close and were regarded as the same because both solvents were ddH_2_O. The quantum yields of the fluorescent sample (Au NCs) were calculated by Equation (1).

## 3. Results

### 3.1. Characterization of Silver Nanoparticles

In a previous report, it was demonstrated that AgNPs play important roles in the synthesis of Au NCs using small molecules as capping ligands [20]. However, differently shaped AgNPs could exhibit capping ligands with different binding abilities. In this study, four different shapes of AgNPs were synthesized: quasi-spherical silver nanoparticles (QS-AgNPs), spherical silver nanoparticles (S-AgNPs), triangular silver nanoplates (T-AgNPts), and nanoplates (AgNPts). After AgNP synthesis, the UV-Visible spectrophotometer was employed to examine whether the SPR absorption band was observed and located at its characteristic wavelengths. Figure 1A–D depict the UV-Vis spectra and TEM images of the as-prepared colloidal silver nanoparticles with different shapes. QS-AgNPs, shown in Figure 1A, exhibit a symmetric LSPR band that has the highest absorption at approximately 408 nm. Silver nanoplates synthesized using a three-stage photochemical reaction (Figure 1B) show bands peaking at 672 nm and 343 nm, correctly corresponding to the in-plane dipolar LSPR and out-of-plane quadrupole modes, respectively. The TEM image showed that the colloids consist of triangular- and round-shaped AgNPts. Triangular silver nanoplates were prepared using the plasmon-mediated process involved in the irradiation of the sodium lamp. As shown in Figure 1C, the UV-Vis spectra of T-AgNPts showed two characteristic bands peaking at 702 nm and 340 nm, corresponding to the in-plane dipolar LSPR mode and out-of-plane quadrupole mode, respectively. Spherical silver nanoparticles (S-AgNPs), shown in Figure 1D, exhibit a symmetric LSPR band that peaks at around 408 nm, similar to the UV-Vis absorption of quasi-spherical silver nanoparticles. The TEM image exhibits the round-shaped morphology of the colloids, which could be either spherical or plated. However, since no peak was detected at 330–340 nm in the UV-Vis spectra, the products are indeed spherical AgNPs.

### 3.2. Synthesis of Gold Nanoclusters Mediated by Silver Nanoparticles

A bottom-up (atoms to clusters) strategy was employed to synthesize Au NCs with small molecules as capping ligands. Three capping ligands (DTT, oxidized DTT, and GSSG) and four different shapes of as-prepared AgNPs were used in the synthesis of Au NCs. In general, three required components have to be present in the Au NC synthesis solution: Au^3+^, Au^3+^-reducing agents (AgNPs), and capping ligands. In the typical synthesis of Au NCs, AgNP colloids, HAuCl_4_, and capping ligands are sequentially placed in the reaction solution and mixed. The mixture is then incubated at 37 °C. Figure 2A (odd numbers for QS-AgNPs, even numbers for T-AgNPts) and 2B (odd numbers for AgNPts, even numbers for S-AgNPs) show the results for when GSSG molecules were used as capping ligands and all four different shapes of AgNPs were used as reducing agents. After several hours of incubation, the reaction solution gradually turned bright red under the irradiation of the UV-box equipped with dual wavelengths, 254 and 365 nm. These results suggest that all four different shapes of AgNPs could be used to successfully synthesize Au NCs. To examine whether different concentrations of AgNPs influence the synthesis of Au NCs, different concentrations of QS-AgNPs (0.0, 3.2, 4.8, 6.4, and 8.0 mM) were used in the synthesis. All of the results, except those for 0.0 mM, showed a bright red color under UV-Box when 0.4 mM HAuCl_4_ and 5 mM GSSG were used. The synthesized GSSG-Au NCs (labelled temporarily) were further investigated to determine their maximal excitation and emission wavelengths using fluorescence spectroscopy. All four synthesized GSSG-Au NCs mediated by four different shapes of AgNPs showed the same maximal excitation and emission wavelengths, at 354 and 623 nm, respectively. In the absence of GSSG, no fluorescent emission was detected (Figure 2A,B, lanes 1 and 2), which revealed that capping ligands are required to produce Au NCs.

To verify whether all four different shapes of Ag NPs are effective for the synthesis of Au NCs with other capping ligands, DTT was used to replace GSSG. The as-prepared reaction solutions of DTT-Au NCs (labeled temporarily), under the detection of the UV-box, are shown in Figure 3A,B. Interestingly, the results clearly show that DTT-Au NCs can be successfully synthesized when QS-AgNPs (#3, 5, 7, and 9 of Figure 3A) and AgNPts (#4, 6, 8, and 10 of Figure 3B) are used in the reaction. In contrast, much less red fluorescence was observed from the solutions when T-AgNPts (#4, 6, 8, and 10 of Figure 3A) and S-AgNPs (#3, 5, 7, and 9 of Figure 3A,B) were used in the synthesis of DTT-Au NCs, indicating that T-AgNPs and S-AgNPs exhibit less efficiency during the synthesis of DTT-Au NCs. To further understand whether more capping ligands or AgNPs are required to optimize the synthesis of Au NCs, higher concentrations of DTT and AgNPs were used to synthesize DTT-Au NCs. However, fluorescent intensities did not increase significantly, suggesting that higher concentrations of capping ligands or AgNPs are not able to increase the percent yield of Au NCs.

We further used an oxidized form of DTT (DTTox) to synthesize DTTox-Au NCs (labeled temporarily). However, the results are very similar to those using DTT as the capping ligands (Figure 4). Therefore, DTTox-Au NCs can be successfully synthesized when QS-AgNPs and AgNPts are used in the reaction. Less efficient Au NC synthesis was observed when S-AgNPs or T-AgNPs were used to synthesize DTT-Au NCs or DTTox-Au NCs. The difference in efficiency of Au NC synthesis could be attributed to the discrepancy in the association of AgNPs with capping ligands.

It is conceivable that the association of capping ligands with these AgNPs could play an important role in the synthesis of Au NCs. To evaluate whether GSSG, DTT, and DTTox can associate with AgNPs, differently shaped AgNPs were mixed with these capping ligands. The mixtures were detected by UV-Vis spectrophotometry. Figure 5 demonstrates the time course of the UV-Vis spectra after the DTT molecules were mixed with AgNPs. All of the maximal absorption wavelengths were red-shifted after DTT was associated with various AgNPs, indicating that aggregates of AgNPs were formed. In addition, the intensities decreased significantly. To confirm the aggregation of AgNPs, TEM images (upper panels of Figure 5A,D) of AgNPs interacting with DTT were also taken. The images show that QS-AgNPs (upper panel of Figure 5A) aggregated slightly after interacting with DTT, while T-AgNPs (upper panel of Figure 5D) aggregated aggressively after interacting with DTT. To further understand the kinetics of AgNP aggregation, the figures were replotted to show red-shift wavelength vs. interaction times. The aggregation time can be estimated from the new figures (Figure 5E–H). The results show that the extent of the red-shift wavelengths differs. To investigate the aggregation kinetics of AgNPs in the presence of DTT, the red-shift curves of the samples were analyzed. Exponential functions were employed to fit the curves. The fitting parameters are shown in the panel of each figure. The calculated aggregation times were 213 min for QS-AgNPs (highlighted in red), 18.4 min for S-AgNPs, 166 min for AgNPts, and 22.1 min for T-AgNPts. The aggregation time of QS-AgNPs was longer than that of T-AgNPts, consistent with the observations of the TEM images shown in Figure 5A,D. The aggregation time of S-AgNPs was also much shorter than that of the QS-AgNPs. After comparing the aggregation time with the results of Au NC synthesis, it was found that the longer the aggregations time, the better the Au NC synthesis. These data suggest that the synthesis of Au NCs is dependent on the aggregation rate of AgNPs after being mixed with capping ligands.

Based on the current data and results shown for the as-prepared Au NCs, it is concluded that QS-AgNPs and AgNPts are the best silver nanoparticles for mediating the synthesis of Au NCs for all three small molecules as the capping ligands (Table 1). “Good” or “fair” is classified based on whether the as-prepared Au NCs solution can be loaded onto the electrophoretic mobility shift assay (EMSA) and detected by the UV-box without further concentrating the process. In this study, GSSG molecules could be used as capping ligands to synthesize GSSG-Au NCs no matter what shape of AgNPs were used for synthesis.

### 3.3. Characterization of Gold Nanoclusters

To verify that Au NCs had been successfully synthesized in the presence of AgNPs, the as-prepared Au NCs were observed under TEM. The TEM images showed that synthesized particles were approximately 5–6 nm or smaller (Figure 6A–D). The size of the DTT-Au NCs particles was less than 5 nm when QS-AgNPs or AgNPts were used to mediate the synthesis. However, the average size of the GSSG-Au NC particles was approximately 5–6 nm. These results confirm that the presented method can successfully synthesize Au NCs. To confirm that the images of the particles detected by TEM were not smaller-sized AgNPs, an energy dispersive X-ray spectrometer (EDS) was employed to analyze the compositions of the particles. The EDS results for DTT-Au NCs synthesized by QS-AgNPs and GSSG-Au NCs synthesized by S-AgNPs are shown in Figure 7A,B. The EDS data show that the major components of the synthesized particles were Au, C, and S in both cases. These results suggest that Au is from the core of the cluster, and S is from the capping ligands (DTT and GSSG molecules). No silver elements were detected in these EDS analyses. Therefore, these data confirm that the particles are indeed DTT-Au NCs and GSSG-Au NCs, respectively.

A previous report suggested that limited ions have to be entrapped in a sequestered space during the synthesis of nanoclusters [12]. In this study, AgNPs were employed as a reaction platform, providing surfaces that play dual roles in the synthesis, as reducing agents and attracting capping ligands on its surfaces to form small spaces to accommodate limited Au atoms. Figure 8A,C show the TEM images of DTTox-Au NCs, which were synthesized in the early stage of the synthesis (5 h only). Figure 8A demonstrates that the newly synthesized DTTox-Au NCs, which were reduced by QS-AgNPs, were arranged into a round shape, similar to that of the QS-AgNPs. Figure 8C demonstrates that the newly synthesized DTTox-Au NCs, which were reduced by AgNPts (mixtures of round and triangular nanoplates), were arranged into circular and triangular shapes, similar to the outline of AgNPts. These data indicate that Au NCs were indeed synthesized on the surface of AgNPs. The EDS data show that the major metal elements were Au and Cu, and very little Ag was detected (Figure 8B,D). These results reveal that the particles were Au NCs. It has to be noted that hollow spaces without Au NCs in the center can be observed in Figure 8C, indicating that capping ligands cannot easily reach the center of triangular or circular AgNPts. On the contrary, DTTox-Au NCs filled in the whole space when QS-AgNPs were used.

To evaluate the properties of these Au NCs, some of the Au NC products were measured to determine their fluorescent wavelengths of excitation and emission (Table 2). Fluorescent quantum yields were also calculated (Table 3). As the data revealed, the synthesized GSSG-Au NCs showed the same emission wavelength at 623 nm, and DTT-Au NCs emitted the highest fluorescent intensity at 640 nm. These results reveal that the fluorescent emission wavelength is related to the capping ligands. The fluorescent quantum yields showed a slight difference, which could have been due to the standard error.

## 4. Conclusions

We previously reported a facile synthesis method for thiolated Au NCs, mediated by quasi-spherical AgNPs. It was suggested that QS-AgNPs provide the surface of its particles and play two important roles in the synthesis of Au NCs. One is to reduce gold ions, and the other is to create a suitable space with small capping ligands for the synthesis of clusters. Au ions distributed on the surface of AgNPs are trapped and reduced by the AgNPs while the capping ligands are attracted to the surface of the AgNPs, resulting in an association with the reduced Au atoms. Therefore, it is conceivable that the shape and size distributions of AgNPs could be critical for the binding of capping ligands and dictate the successfulness of the synthesis. In this study, AgNPs with four different morphologies (QS-AgNPs, S-AgNPs, T-AgNPts, and AgNPts) were prepared and used to examine whether this approach is dependent on the morphology of AgNPs. Based on the data shown, the results are distinguishable, and QS-AgNPs and AgNPts are the best choices to mediate the synthesis of Au NCs for all three tested small molecules as the capping ligands. The TEM images showed that Au NCs are synthesized on the surface of silver nanoparticles (Figure 8). Differently shaped AgNPs serve as platforms to synthesize Au NCs with small capping ligands (Figure 9). It was demonstrated that Au NCs were transiently assembled on the surface of silver nanoparticles in the method. The formation of Au NCs was observed on the whole surface of the QS-AgNPs if the synthesis of DTTox-Au NCs was mediated by QS-AgNPs (Figure 9A). In contrast, the formation of Au NCs was only observed on the edges and corners of AgNPts if the synthesis of DTTox-Au NCs was mediated by AgNPts. Our data imply that DTT and/or gold ions cannot bind to or reach the center of T-AgNPts (Figure 9B). The synthesized Au NCs display similar fluorescent properties, including quantum yields and excitation and emission wavelengths. However, the size distributions of silver nanoparticles were not discussed in this study, because the synthesis of uniformly sized AgNPs is beyond our capability.

## Figures and Tables

**Figure 1 nanomaterials-13-02338-f001:**
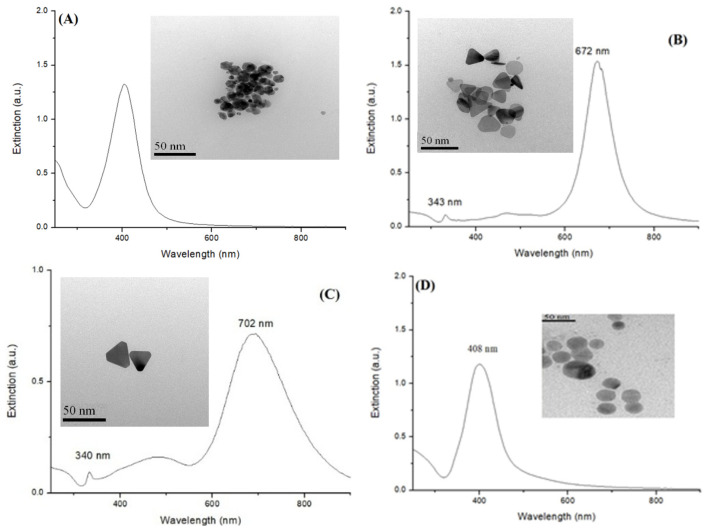
UV-Vis spectra and TEM images of colloidal silver nanoparticles with different shapes. (**A**) Quasi-spherical silver nanoparticles (QS-AgNPs), (**B**) silver nanoplates (AgNPts), (**C**) triangular silver nanoplates (T-AgNPts) and (**D**) spherical silver nanoparticles (S-AgNPs).

**Figure 2 nanomaterials-13-02338-f002:**
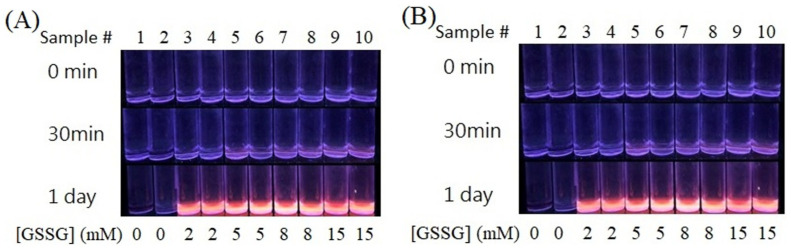
Synthesis of GSSG-Au NCs using 0.4 mM HAuCl_4_ and different shapes of 4.8 × 10^−2^ mM AgNPs. Concentrations of GSSG are indicated at the bottom of each figure. The images of the as-prepared GSSG-Au NCs solutions were taken under UV-box detection. Sample # (numbers) are indicated above. (**A**) Odd numbers (3, 5, 7, and 9) indicate that QS-AgNPs were used as the reductants and even numbers (4, 6, 8, and 10) indicate that T-AgNPts were used as the reductants. (**B**) Odd numbers indicate that AgNPts were used as the reductants and even numbers indicate that S-AgNPs were used as the reductants.

**Figure 3 nanomaterials-13-02338-f003:**
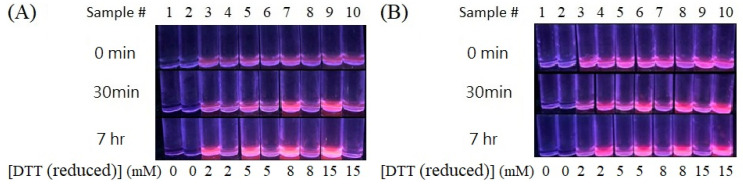
Synthesis of DTT-Au NCs using 0.4 mM HAuCl_4_ and different shapes of 4.8 × 10^−2^ mM AgNPs. Concentrations of DTT are indicated on the bottom of each figure. The images of aqueous solutions of the as-prepared DTT-Au NCs are taken under UV-box detection. Sample # (numbers) are indicated above. (**A**) Odd numbers (3, 5, 7, and 9) indicate that QS-AgNPs were used as the reductants and even numbers (4, 6, 8, and 10) indicate that T-AgNPs were used as the reductants. (**B**) Odd numbers indicate S-AgNPs were used as the reductants and even numbers indicate that AgNPts were used as the reductants.

**Figure 4 nanomaterials-13-02338-f004:**
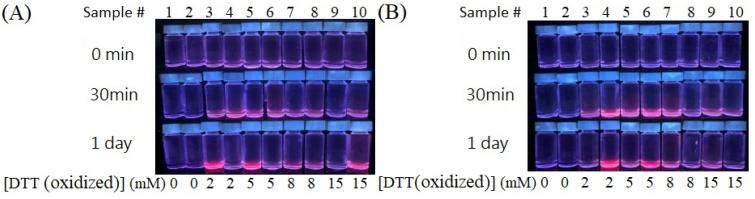
Synthesis of DTTox-Au NCs using 0.4 mM HAuCl_4_ and different shapes of 4.8 × 10^−2^ mM AgNPs. Concentrations of DTTox are indicated on the bottom of each figure. The images of aqueous solutions of the as-prepared DTTox-Au NCs are shown under UV-box detection. Sample # (numbers) are indicated above. (**A**) Odd numbers (3, 5, 7, and 9) indicate that QS-AgNPs were used as the reductants and even numbers (4, 6, 8, and 10) indicate that T-AgNPs were used as the reductants. (**B**) Odd numbers indicate S-AgNPs were used as the reductants and even numbers indicate that AgNPts were used as the reductants.

**Figure 5 nanomaterials-13-02338-f005:**
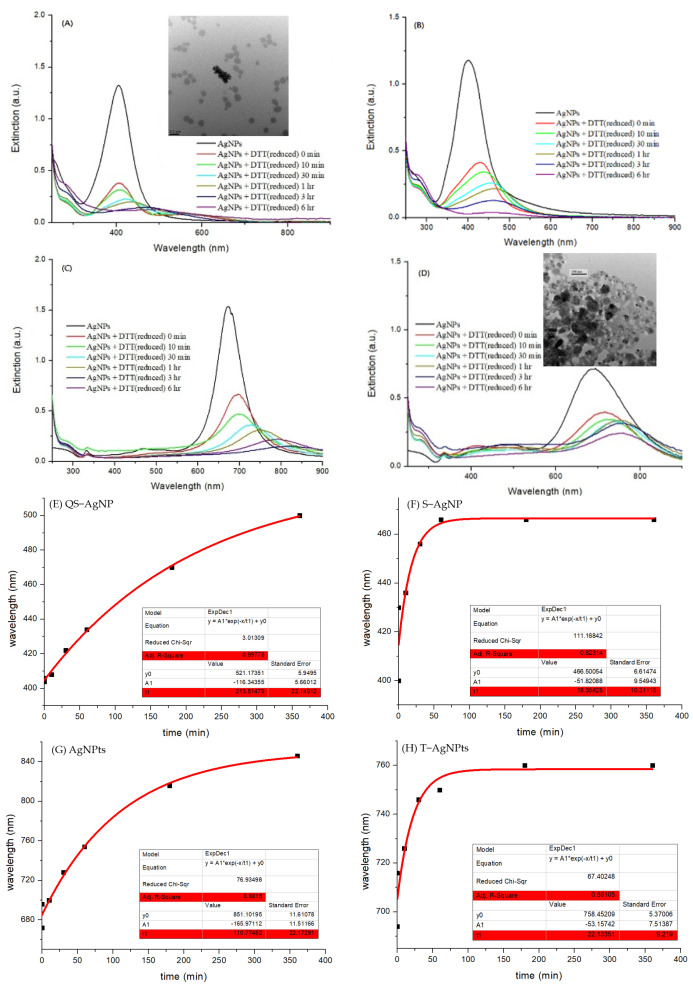
UV-Vis spectra of the association of AgNPs with DTT (**A**–**D**) and re-plots of red-shift wavelengths versus interaction times (**E**–**H**). (**A**) QS-AgNP colloids, (**B**) S-AgNP colloids, (**C**) AgNPts colloids, and (**D**) T-AgNPts colloids at various interaction times. Time-dependent AgNP aggregations were shown in (**E**) QS-AgNP colloids, (**F**) S-AgNP colloids, (**G**) AgNPts colloids and (**H**) T-AgNPts colloids.

**Figure 6 nanomaterials-13-02338-f006:**
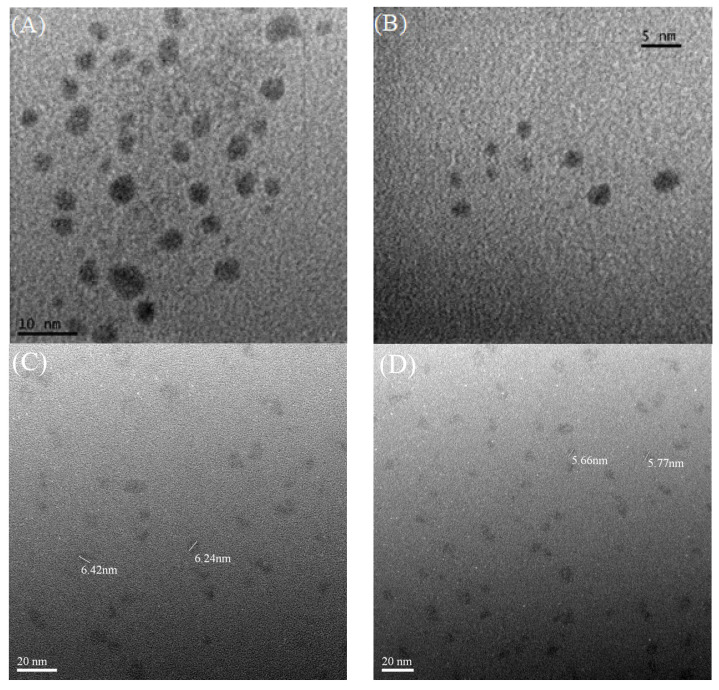
TEM image of Au NCs. (**A**) DTT-Au NCs synthesized by QS-AgNPs. The sizes of the particles are under 5 nm. (**B**) DTT-Au NCs synthesized by AgNPts. The sizes of the particles are under 5 nm. (**C**) GSSG-Au NCs synthesized by S-AgNPs. Most of the particles are approximately 6 nm in size. (**D**) GSSG-Au NCs synthesized by T-AgNPts. Most of the particles are approximately 5–6 nm in size.

**Figure 7 nanomaterials-13-02338-f007:**
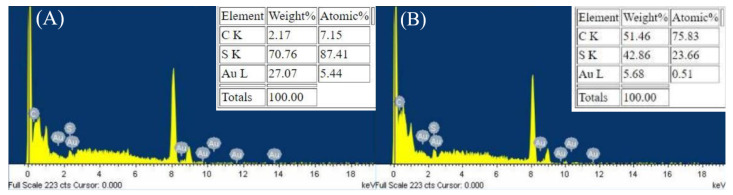
Energy dispersive X-ray spectrometer (EDS). (**A**) EDS analysis of DTT-Au NCs synthesized by QS-AgNPs. (**B**) EDS analysis of GSSG-Au NCs synthesized by S-AgNPs. In both cases, the analyzed particles consist of C, S, and Au elements (upper panel in **A**,**B**). No Ag elements were detected.

**Figure 8 nanomaterials-13-02338-f008:**
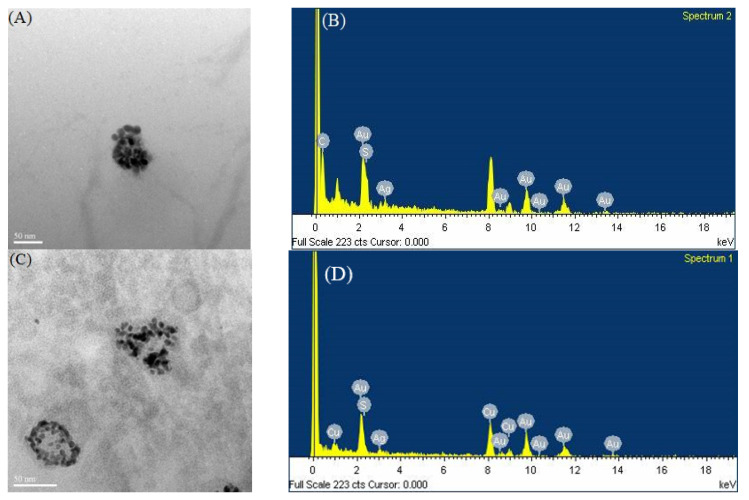
TEM image of DTTox-Au NCs synthesized for 5 h. (**A**) DTTox-Au NCs synthesized by QS-AgNPs. The particles were less than 5 nm in size. (**B**) The EDS analysis shows very little Ag. (**C**) DTTox-Au NCs synthesized by AgNPts. The sizes of the particles are under 5 nm. (**D**) EDS analysis.

**Figure 9 nanomaterials-13-02338-f009:**
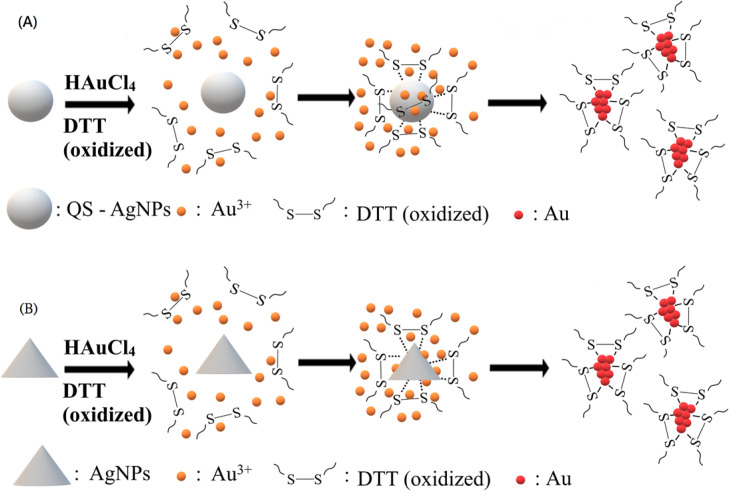
Two differently shaped AgNPs serve as a platform to synthesize Au NCs with small capping ligands. (**A**) The synthesis of DTTox-Au NCs mediated by QS-AgNPs. The formation of Au NCs was observed on the whole surface of the QS-AgNPs. (**B**) The synthesis of DTTox-Au NCs mediated by AgNPts. The formation of Au NCs was only observed on the edges and corners of the AgNPts.

**Table 1 nanomaterials-13-02338-t001:** Synthesis efficiency of Au NCs using various AgNPs and capping ligands.

Capping Ligands	QS-AgNPs	T-AgNPts	AgNPts	S-AgNPs
DTTox	good	fair	good	fair
DTT	good	fair	good	fair
GSSG	good	good	good	good

**Table 2 nanomaterials-13-02338-t002:** Excitation and emission wavelengths of the synthesized Au NCs.

	QS-AgNPs		AgNPts	
Capping Ligands	Excited Wavelength	Emission Wavelength	Excited Wavelength	Emission Wavelength
GSSG	354 nm	623 nm	354 nm	623 nm
DTT	354 nm	640 nm	354 nm	643 nm
DTTox	360 nm	676 nm	357 nm	668 nm

**Table 3 nanomaterials-13-02338-t003:** Quantum yields of the synthesized Au NCs.

Capping Ligands	DTT	DTT	GSSG	GSSG	GSSG	GSSG
AgNPs	QS-AgNP	AgNPt	QS-AgNP	T-AgNPt	S-AgNP	AgNPt
QY(%)	7.2	5.4	6.8	6.3	7.3	7.8

## Data Availability

All data generated and analyzed during this study are included in this paper and the attached Appendix A. The data presented in this study are available on request from the corresponding author.

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
