# Peer review of "Shape Dependence of Silver-Nanoparticle-Mediated Synthesis of Gold Nanoclusters with Small Molecules as Capping Ligands"

_nanomaterials, 2023, doi:10.3390/nano13162338_

Round 1

Reviewer 1 Report

The manuscript “Shape-dependent of silver nanoparticles-mediated synthesis of gold nanoclusters with small molecules as capping ligands” describes an approach for gold nanoclusters synthesis using differently shaped silver nanoparticles and small capping ligands.

In my opinion, the manuscript has several major issues that have to be addressed before it can be considered for publication.

1)    The method described has already been reported by the authors in the paper “Silver Nanoparticles-Mediated Synthesis of Fluorescent Thiolated Gold Nanoclusters”, published in Nanomaterials, 11 (2021) 2835. The significance of the present study is not sufficiently highlighted, and no application for AuNCs is reported

2)    The authors are advised to report the minimum and maximum concentration of AgNPs required for the AuNCs synthesis and if the AgNPs concentration influences the final concentration and properties of AuNCs

3)    Given the remarkably comparable shapes of the quasi-spherical and spherical AgNPs, could the authors hypothesize why there is such a major difference between AuNCs synthesized from them? Is the size of AgNPs a decisive factor for AuNCs synthesis?

The manuscript is well written, only minor corrections are needed

Reviewer 2 Report

The article is interesting, but authors need to improve it to match the level of the Nanomaterials.

- In the Materials section please add the data on the purity of the reagents used.

- I recommend to add computer simulations of the Ag NPs abs spectra (Fig. 1; Fig. 5) in order to match it with the experimental spectra.

- In the Table 1 you use "good/fair" scale for the evaluation of the synthesis efficiency of Au NCs. I strongly recommend to use quantitative rather than qualitative assessment scale. Also please clarify why the value of 15,000 cps/sec was used as a threshold?

- The size of the obtained Au NCs are about 5-6 nm. Such particles are closer to nanoparticles rather that nanoclusters. Moreover such particles should have a strong SPR (Au nanoparticles of the same size precipitated in glass posses SPR band in the visible range, e.g. check Ref. Shakhgildyan, G. Yu, et al. "Thermally-induced precipitation of gold nanoparticles in phosphate glass: effect on the optical properties of Er3+ ions." Journal of Non-Crystalline Solids 550 (2020): 120408. or Sigaev, Vladimir N., et al. "Spatially selective Au nanoparticle growth in laser-quality glass controlled by UV-induced phosphate-chain cross-linkage." Nanotechnology 24.22 (2013): 225302). So I strongly ask you to provide the UV-Vis spectra of the obtained Au NCs in order to check presence/absence of the SPR band. These spectra could be used for evaluation of the nature of the synthetized NCs or NPs.

- Please clarify the scheme on Fig. 9. It seems that the final parts of the scheme (i.e. formation of Au NC) are the same in both a) and b) panels. But in the caption (and in the text) you stated that results are different. Also you need to add short description for each step (i.e. arrow).

- There is a lack of the Conclusions section in the article. Please add it.

Round 2

Reviewer 1 Report

Accept in present form

it is ok

Reviewer 2 Report

All comments were addressed. I recommend to accept the article.